# *Dendropanax morbifera* Leaf Polyphenolic Compounds: Optimal Extraction Using the Response Surface Method and Their Protective Effects against Alcohol-Induced Liver Damage

**DOI:** 10.3390/antiox9020120

**Published:** 2020-02-01

**Authors:** Taekil Eom, Kyeoung Cheol Kim, Ju-Sung Kim

**Affiliations:** 1Subtropical/Tropical Organism Gene Bank, SARI, Jeju National University, Jeju 63243, Korea; taekil7@hanmail.net; 2Majors in Plant Resource and Environment, College of Agriculture & Life Sciences, SARI, Jeju National University, Jeju 63243, Korea; cheolst@jejunu.ac.kr

**Keywords:** *Dendropanax morbifera*, response surface method, antioxidant, alcoholic liver

## Abstract

The response surface methodology was used to optimally extract the antioxidant substances from *Dendropanax morbifera* leaves. The central composite design was used to optimally analyze the effects of ethanol concentration, sample to solvent ratio, extraction temperature, and extraction time on the total flavonoids (TF) content, ferric reducing antioxidant power (FRAP), and Trolox equivalent antioxidant capacity (TEAC). All three parameters were largely influenced by the ethanol concentration and extraction temperature, while TEAC was also influenced by the sample to solvent ratio. The maximum values of TF content, FRAP, and TEAC were achieved under the following extraction conditions: 70% ethanol, 1:10 sample to solvent ratio, 80 °C, and 14 h. The *D. morbifera* leaf extracts (DMLE) produced under these optimum extraction conditions were investigated to determine their preventive effects on alcohol-induced liver injury. The DMLE was shown to prevent liver injury by scavenging the reactive oxygen species generated by alcohol. In addition, composition analysis of DMLE found high contents of chlorogenic acid and rutin that were determined to inhibit alcoholic liver injury. The findings of this study suggest that DMLE could prove useful as a functional food product supplement to prevent liver injury caused by excessive alcohol consumption.

## 1. Introduction

The recent global economic growth has led to a steep rise in alcohol consumption and increasing concerns for various health and social issues. Alcohol in the human body is first converted to acetaldehyde via the oxidation process, mediated by alcohol dehydrogenase (ADH) in the liver, and then converted to acetic acid via the same process, but mediated by aldehyde dehydrogenase (ALDH) [1]. The oxidation pathway used to metabolize alcohol in the liver is largely governed by three enzymes: ADH, cytochrome P450 2E1 (CYP2E1), and catalase (CAT) [2]. The reactive oxygen species (ROS) generated from excessive alcohol consumption are known to oxidize intracellular macromolecules, such as DNA, proteins, and lipids, and induce oxidative stress in the liver, causing functional damages. If the liver injury persists, the condition advances to alcoholic fatty liver disease [3,4]. A large number of recent studies have focused on developing materials from various medicinal plants that can prevent alcoholic liver injury without side effects [5,6].

In Korea, the native plants of the *Araliaceae* family have historically been used as medicinal herbs. These species include *Aralia cordata*, *Aralia elata*, *Dendropanax morbifera*, *Eleutherococcus senticosus*, *Fatsia japonica*, *Kalopanax septemlobus*, and *Panax ginseng*. In most cases, the root or the bark is used as the material for medicinal effects [7]. Specifically, Dendropanax morbifera is a subtropical evergreen tree species that is also distributed in other regions of North-East Asia, such as China and Japan. Previous studies have shown that *D. morbifera* leaf extracts (DMLE) are enriched with polyphenolic compounds that offer a diverse array of functionalities, including antioxidant, anti-inflammatory, anti-obesity, and anti-cancer activities, as well as immunoregulatory functions [8,9,10,11].

The extraction of functional substances and their application as functional materials from medicinal plant resources generally involves solvents, such as water and ethanol. However, certain factors, including solvent composition, extraction temperature, and extraction time, can influence extraction yields; therefore, such factors require optimization [12]. The most important goal of any experimental design is to locate the conditions of independent variables to optimize either the maximum or minimum values for the response of interest. When a polynomial equation is derived using the response variables, depending on the experimental conditions of independent variables, and then plotted on coordinates, the line or plane crossed by the estimated values is referred to as the response surface. The method that searches for the optimum condition within the interests of a researcher is known as the response surface methodology (RSM). After a variance analysis or a factorial experiment have determined the potential optimal conditions, RSM is used to analyze the relationship among the independent variables and response variables that constitute the optimum conditions [13]. RSM has thus been used in various studies on food products to optimize the food production process [14]. RSM has also been applied to the design and optimization of experiments involving medicinal plants and the extraction of a variety of functional substances; specifically regarding the analysis of the relationships among independent variables and response variables, and the optimization of extraction conditions [15].

This study aims to evaluate the optimal extraction conditions for DMLE by applying RSM based on four independent variables (ethanol concentration, sample to solvent ratio, extraction temperature, and extraction time). This study also verifies the effects of DMLE on alcoholic liver injury to suggest its potential use as a supplement in functional food products to prevent alcoholic liver injury.

## 2. Materials and Methods

### 2.1. Chemicals and Reagents

The high-performance liquid chromatography (HPLC) standards (chlorogenic acid, rutin) were purchased from Chemfaces (Wuhan, China). Antioxidant assay reagents: 2,2′-azino-bis(3-ethylbenzothiazoline-6-sulfonic acid) diammonium salt (ABTS), 2′,7′-dichlorofluorescin diacetate (DCFH-DA), trolox, 2,4,6-tris(2-pyridyl)-s-triazine (TPTZ), quercetin, aluminum chloride hexahydrate (AlCl_3_·6H_2_O), potassium persulfate, and propidium iodide (PI) were purchased from Sigma-Aldrich (St. Louis, MO, USA). Cell culture reagents: Dulbecco’s modification of Eagle’s medium (DMEM), penicillin/streptomycin antibiotic mix, glutamine, and fetal bovine serum (FBS) were purchased from Welgene (Gyeongsan, Korea). All other chemicals used were of 99% purity or higher.

### 2.2. Plant Materials and Extraction

Fresh *D morbifera* leaves were supplied by Jeju Dendropanax Co. Ltd., (Jeju, Korea). The leaves were washed using tap water and dried at room temperature for 1 week. Dried plant samples were ground using a blender and sieved to a particle size of 100 μm. Sample powder (1 g) was extracted using the extract conditions designed for the RSM. After extraction, the extracts were filtered with Whatman No. 2 filtering paper (Whatman International Limited, Kent, England) and dried using a rotary vacuum evaporator (Hei-VAP Precision, Heidolph, Germany) and stored at −20 °C until further analysis.

### 2.3. RSM Experimental Design

RSM was used to establish the optimum extraction conditions for DMLE. A central composite design was applied based on the results of a preliminary study. The ranges of optimal conditions for each leaf sample were determined with regards to the independent variables (X_n_) of the extraction process: the ethanol concentration (X_1_), sample to solvent ratio (X_2_), extraction temperature (X_3_), and extraction time (X_4_), and then coded into three levels (−1, 0, 1) (Table 1). Under the influence of the independent variables (as described above), the dependent variables—the total flavonoids (TF) content, Trolox equivalent antioxidant capacity (TEAC), and ferric reducing antioxidant power (FRAP)—were measured in triplicate, and the data analyzed using regression analysis. The MiniTab Ver 17 (Minitab Inc., IL, State College, PA, USA) program was used to predict the optimal conditions for the regression analysis. In the quadratic regression analysis of the data, the response function (Y) was defined as shown below, which includes the linear, quadratic, and interacting components as well as the suggested model:(1)$$Y=\beta_{0}+\sum_{i=1}^{k}\beta iXi+\sum_{i=1}^{k}\beta iiXi^{2}+\sum_{i\neq j=0}^{k}\beta ijXiXj$$

*Y* represents the dependent variables, *β**_0_* is a constant coefficient, and *k* is a test variable. *βi, βii,* and *βij* are the regression coefficients for the intercept, linear, and quadratic equations, respectively. The contour map for each dependent variable was used in the analysis to monitor the extraction profile and predict the range of optimal conditions.

### 2.4. Total Flavonoids (TF) Content

The TF content was measured, as described by Kim et al. (2012) [16]. Briefly, the following were added to 100 μL of DMLE: 300 μL of ethanol, 20 μL of 10% aluminum chloride, 20 μL of 1 M potassium acetate, and 560 μL of distilled water. The mixture was left to react for 1 h. Optical density (OD) was then measured at 415 nm using a microplate reader and expressed as a quercetin equivalent (QE) through the standard curve drawn using quercetin (*Y* = 0.003*x* + 0.049, *R^2^*: 0.999).

### 2.5. Trolox Equivalent Antioxidant Capacity (TEAC)

The TEAC was measured, as described by Zulueta et al. (2009) with modifications [17]. Briefly, 7 mM ABTS and 2.45 mM potassium persulfate were mixed and left to react in a dark room. The resulting ABTS radicals were calibrated to display a mean standard deviation (SD) OD of 0.70 ± 0.02 at 734 nm prior to use in the experiments. After adding 50 μL of DMLE to 1 mL of ABTS, the reaction was left for 5 min, after which the OD was measured at 734 nm using a UV-spectrometer (UV-1800, Shimadzu, Japan). The standard curve was drawn using Trolox, wherein the Trolox concentration (mM Trolox equivalent [TE]/g) was expressed per gram of DMLE (Y = 0.679*x* − 0.001, *R^2^*: 0.999).

### 2.6. Ferric Reducing Antioxidant Power (FRAP)

The FRAP was measured following Benzie and Strain (1996) [18]. To prepare the FRAP working solution, 0.3 M sodium acetate buffer (pH 3.6), 20 mM FeCl_3_, and 10 mM TPTZ were mixed in a 10:1:1 ratio, and left to react for 30 min at 37 °C prior to use. After adding 150 μL of the FRAP working solution to 50 μL of DMLE, the reaction was left for 15 min at 37 °C. The OD was then measured at 595 nm using a microplate reader. The standard curve was drawn using FeSO_4_, wherein the FeSO_4_ reducing power (mM FeSO_4_/g) was expressed per gram of DMLE (*Y* = 0.045*x* + 0.0003, *R^2^*: 0.999).

### 2.7. Cell Culture and Cytotoxicity Determination Using an MTT Assay

The human hepatoma cancer cell line HepG2 was obtained from the American Type Culture Collection (Manassas, VA, USA). HepG2 cell lines were cultured and maintained in DMEM supplemented with 100 U/mL penicillin, 100 μg/mL streptomycin, 2 mM glutamine, and 10% FBS at 37 °C under a humidified atmosphere with 5% CO_2_.

Cells were subcultured for 24 h in 96-well plates at a density of 5 × 10^4^ cells/well in DMEM supplemented with 10% FBS. After 24 h, various concentrations of DMLE were treated and the cytotoxicity was measured using an MTT assay to measure the succinate dehydrogenase activity in mitochondria. The culture media were aspirated and the MTT solutions (final concentration, 1 mg/mL) were added into each well and incubated for 4 h. The media were aspirated again and the purple color crystals were dissolved with DMSO. The absorbance in each well was measured at 540 nm using an i3 multifunctional microplate reader (Molecular Devices, San Jose, CA, USA).

### 2.8. Apoptosis Analysis using Flow Cytometry

Apoptotic cells were detected by flow cytometry using the PI method. Briefly, cells were incubated with ethanol and the DMLEs were harvested and were washed once with PBS. The cells were fixed with 70% ethanol for 2 h at −20 °C and washed once with PBS. The cells were then resuspended in 300 μL of PBS containing 0.5 mg/mL RNaseA and 0.1 mg/mL PI. After further incubation for 30 min in the dark, flow cytometry was performed with an LSR Fortesa flow cytometer (Becton Dickinson, San Jose, CA, USA). PI fluorescence intensity was collected through an FL-2 filter (585 nm). Events within the linear FL-2 area vs. FL-2 width plot were used in the analyses. The population of apoptotic cells was obtained from the sub-G_1_ population in the FL2-A histogram.

### 2.9. Cellular ROS Measurement using Flow Cytometry

The redox-sensitive fluorescent probe DCFH-DA was used to detect the production of intracellular ROS. Cells were treated with various ethanol concentrations for 12 h, then harvested, washed, and re-suspended in 10 μM DCFH-DA solutions. The cells were then incubated at 37 °C for 30 min in a dark room before being analyzed by a flow cytometer. The values were expressed as a percentage of fluorescence intensity relative to the blank control cells.

### 2.10. HPLC Q-TOF Mass Analysis

The phytochemicals in the DMLE were analyzed using HPLC-Quadruple Time of Flight (Q-TOF) mass spectrophotometry. The HPLC system (Agilent infinity 1260 series, Germany), with an incorporated photodiode array detector (DAD) and an Impact HD Q-TOF mass spectrometer (Bruker Daltonik GmbH, Germany), was equipped with an ElectroESI source that operated on the positive ion mode. A reverse-phase YMC Triat C-18 column (250 × 4.6 mm, 5 μm, YMC Korea) was used at a flow rate of 1 mL/min. The mobile phase consisted of water containing 0.1% TFA (A) and 0.1% TFA containing acetonitrile (B) using the following gradient conditions 0–20 min, 10-25% B; 20–30 min, 25–27.5% B; 30–45 min, 27.5–40% B; 45–50 min, 40–60% B; 50–55 min, 60–100% B; 55–62 min, 100–10% B; and 62–70 min, 10% B. The injection volume was 5 μL. Mass spectra in positive-ion or negative-ion mode were recorded within 70 min. The HPLC profiles of the extracts were measured using a DAD set at four wavelengths of 210, 316, 365, and 520 nm. The analyses were conducted in the positive ion mode in a mass range from *m/z* 50–1000. The ESI source parameters were capillary voltage, 4.5 KV; nebulizing gas pressure, 1.5 Bar; drying gas temperature, 200.0 °C, drying gas flow, 9.0 L/min; funnel 1RF, 250.0 Vpp; transfer time, 50.0 μs; and pre-pulse storage. 2.0 μs. The MS data were analyzed using Data Analysis 4.2 software (Bruker Daltonics, Bremen, Germany).

### 2.11. Chlorogenic Acid and Rutin Quantification in DMLE

Chlorogenic acid and rutin contents in DMLE were analyzed using HPLC coupled with a DAD detection system (Shimadzu Prominence System, Japan). The analysis was performed using a reverse-phase YMC Triat C-18 column (250 × 4.6 mm, 5 μm, YMC Korea). The column temperature was set at 40 °C and the detection wavelength was set at 190–450 nm. The mobile phase consisted of water containing 0.1% TFA (A) and 0.1% TFA containing acetonitrile (B) using the following gradient conditions 0–20 min, 10.25% B; 20–30 min, 25–27.5% B; 30–45 min, 27.5–40% B; 45–50 min, 40–60% B; 50–55 min, 60–100% B; 55–62 min, 100–10% B; and 62–70 min, 10% B. The injection volume was 20 μL. Chlorogenic acid and rutin were quantified using UV absorption at 280 nm.

### 2.12. Statistical Analysis

All experiments were performed at least three times and the results are presented as means ± SD. Statistical comparisons of the mean values were performed using a one-way analysis of variance (ANOVA), followed by Duncan’s multiple range test using Minitab 17 software (Minitab Inc., IL, State College, PA, USA). Differences were considered significant at *p* < 0.05.

## 3. Results and Discussion

### 3.1. Optimized Extraction of D, morbifera Leaf Extracts

To optimize the DMLE extraction process, CCD was used in this study to examine the TF content, TEAC, and FRAP of DMLE. A second-order polynomial model was applied to the experimental data presented in Table 2 to determine the optimal extraction process. The model for the TF content, TEAC, and FRAP data exhibited a high-level fit, with a small standard deviation from each mean value.

The positive linear effects of ethanol concentration and extraction temperature were shown to be significant across all responses, whereas the sample to solvent ratio only displayed a significant effect for the FRAP antioxidant activity. In addition, the second-order effect of ethanol concentration was shown to be significant across all variables, except extraction temperature. The linear effect on extraction time was shown to be insignificant across all parameters: TF content, TEAC, and FRAP. Nonetheless, the second-order effect for extraction time displayed a significant result for TF content.

Two-way interactions for the analysis results of TEAC and FRAP were detected across all experimental variables—ethanol concentration (X_1_), sample to solvent ratio (X_2_), extraction temperature (X_3_), and extraction time (X_4_)—whereas the TF content only showed significant interactions for ethanol concentration and extraction temperature. For a model where one or more parameters can account for the experimental alterations in response variables, the variance analysis results for each response are reflected as a significant *F*-value (Table 3).

The model of variance analysis for testing the data lack-of-fit was shown to be suitable for the experimental data of the response variables TF content, TEAC, and FRAP (*p* < 0.05). A contour graph of the response surface was generated for each response, and a higher level analysis was used to express the interactions among independent variables (Table 3).

### 3.2. Effect of Extraction Variables on TF

The model of TF content produced significant values for the experimental data, and the results of the variance analysis were significant for the linear effects of ethanol concentration (X_1_) and extraction temperature (X_3_), the second-order interaction of ethanol concentration (X_1_^2^) and extraction time (X_4_^2^), and the second-order reciprocal action of ethanol concentration and extraction temperature (X_13_). Based on the *F*-value, ethanol concentration (X_1_) was found to be the most significant factor that increased the TF content of DMLE, which was followed by its second-order interaction (X_1_^2^), extraction temperature (X_3_), reciprocal action (X_13_), and second-order interaction of extraction time (X_4_^2^). The removal of the variables in the variance analysis that had not influenced the model led to the formation of the second-order polynomial equation given below:*Y* = −18.2 + 1.601X_1_ − 0.054X_2_ − 0.402X_3_ + 2.103X_4_ − 0.015X_1_^2^ − 0.091X_4_^2^ + 0.007X_13_(2)

An insignificant *F*-value of the lack-of-fit indicates that the model is suitable for the spatial effects; nonetheless, the reliable predictive value of the model (*R^2^* = 0.846) supports its good predictability for the response variables (Table 3). The contour map of the response surface for the TF content is presented in Figure 1. *X_1_* was found to interact with the sample to solvent ratio, extraction temperature, and extraction time; however, based on the analysis result for the variables, the strongest interaction was found between *X_1_* and *X_3_.* The effect of *X_1_* on the TF content led to the maximum content occurring in the range of 60–80%, while *X_3_* ensured a maximum value at all times at 80 °C, the highest temperature of the response process. These findings are associated with the increased solubility of flavonoid compounds as functional substances contained in plants because the solvent may better penetrate the plant matrix with an increase in the ethanol concentration and the extraction temperature [19,20,21].

In the case of *X_3_*, the value increased with the increase in extraction time, reached a maximum after 10–14 h, and then decreased. This implies that the longer the extraction time, the higher the chance that the flavonoid components can become degraded. It is possible that the decline in the sample to solvent ratio, due to the loss of solvent through vaporization, had a direct influence on the loss of mass transfer during the extraction [22,23]. In contrast, in the case of X_2_, the higher the sample to solvent ratio, the lower the TF content, although the differences were not statistically significant. The results collectively suggested that the optimal conditions to increase the TF content were to have an extraction process with a steady sample to solvent ratio at the highest temperature of 80 °C, with 60–80% ethanol concentration, and for 10–14 h.

### 3.3. Effect of Extraction Variables on FRAP

The results of the FRAP experiments showed similar trends to those seen regarding the TF content. The variance analysis results showed that FRAP activity was influenced by the ethanol concentration (*X_1_*), the linear effect of extraction temperature (*X_3_*), the second-order interaction of ethanol concentration (*X_1_^2^*), the reciprocal action of ethanol concentration and sample to solvent ratio (*X_12_*), the ethanol concentration and extraction temperature (*X_13_*), and the extraction temperature and extraction time (*X_34_*) (Table 2). Among these, *X_1_*, *X_1_^2^*, and *X_13_* showed largest *F*-value regression, and thus stronger dependency. The removal of the factors that did not influence the model led to the formation of the second-order polynomial equation given below:*Y* = 349 + 47.7X_1_ − 6.66X_2_ − 14.38X_3_ + 5.2X_4_ − 0.478X_1_^2^ − 1.623X_4_^2^ + 0.079X_12_ + 0.167X_13_ + 0.235X_14_ + 0.287X_34_(3)

Consistent with the TF content findings, the *F*-value of the lack-of-fit indicated an insignificant spatial effect of the polynomial model; nonetheless, the model exhibited good predictability (*R^2^* = 0.908) (Table 3). The contour map of the response surface for the FRAP activity is presented in Figure 2, and a similar trend to the TF content can be seen. This finding may be explained by the FRAP activity being based on the reducing power of the sample, where the aromatic hydroxyl group in flavonoid compounds acts as an excellent reducing agent. These findings were, therefore, consistent with those of a previous study, in which a correlation between the TF content and FRAP activity was reported [24] The interaction between *X_1_* and *X_2_* or *X_3_* was shown to have a significant effect on the TF content, and although lower values of *X_1_* and *X_2_* resulted in an increase in the TF content, a high value of *X_3_* with a low value of *X_1_* resulted in an increase in the FRAP activity. In addition, the interaction between *X_1_* and *X_4_* was shown to produce a maximum value at 12 h, which is the center of the response model; however, the interactions of *X_12_* and *X_13_* were shown to be stronger. *X_3_* was shown to have a linear effect and the interactions with all other factors, although the interaction with *X_2_* was insignificant. The results collectively suggested that, similar to the TF content, the higher the extraction temperature, the higher the extracted content of TF, while FRAP activity is influenced by the solvent concentration and the extraction time [25].

### 3.4. Effect of Extraction Variables on TEAC

TEAC, as an indicator of antioxidant activity, was estimated by measuring the electron transport ability. As a result, the variance analysis showed that ethanol concentration (*X_1_*), sample to solvent ratio (*X_2_*), and extraction temperature (*X_3_*) had significant linear effects on TEAC values. The reciprocal action of extraction temperature (*X_3_*) and extraction time (*X_4_*) for ethanol concentration (*X_1_*) displayed a stronger influence on TEAC values, but the reciprocal action of *X_1_* and *X_2_* was shown to be insignificant. In contrast, a significant effect on TEAC values was exerted by the reciprocal action of *X_2_* and *X_3_*. As in the case of FRAP, the reciprocal action of extraction temperature and extraction time was found to have had a significant interaction with the TEAC. The removal of the factors that did not influence the model led to the formation of the second-order polynomial equation given below:Y = 144.6 + 6.99X_1_ + 0.769X_2_ − 1.860X_3_ − 10.75X_4_ − 0.075X_1_^2^ + 0.028X_13_ + 0.085X_14_ − 0.019X_23_ + 0.082X_34_(4)

The TEAC data showed a significant lack-of-fit, while the model exhibited a reliable predictability (Table 3). The values of TEAC showed a similar trend to those seen for the TF content and FRAP activity; hence, *X_1_* and *X_3_* were shown to be the most influential factors in all experiments (Figure 3). These findings are consistent with the fact that the aromatic hydroxyl group in flavonoid compounds has an excellent electron transport ability based on the resonance stabilization effect, and support the findings of a previous study, which reported higher antioxidant effects, such as TEAC, for higher TF contents in various medicinal plant extracts [26]. In the case of the TF content and FRAP activity, X_2_ displayed an increasing trend throughout the interactions with the other factors, followed by a decrease, whereas the TEAC values were shown to have a linear response to the other fixed variables. This indicated that although a longer extraction time is disadvantageous for the TF content and FRAP activity, it has beneficial effects on the TEAC values. Therefore, the longer the extraction time, the more substances (other than flavonoids, but with electron transport ability) are extracted. These conclusions are in agreement with the results of a previous study, whereby the extraction time and TEAC were shown to interact with each other [27].

### 3.5. Determination and Validation of the Optimal Conditions

Once the optimal conditions for a given experiment have been determined through the use of response surface analysis, it is vital that the findings be verified using an actual experiment. The analysis results for the extraction of DMLE showed that, among the response variables, ethanol concentration and extraction temperature exerted the largest influence on TF content, TEAC, and FRAP, with relatively less influence from the sample to solvent ratio and extraction time. By optimizing these variables, the extraction conditions that produce maximum values of TF content, TEAC, and FRAP were determined. The optimal conditions, using the MiniTab program and based on the second-order polynomial equations obtained from the experimental data for each factor, was shown to predict the extraction condition that satisfies the maximum values of TF content, TEAC, and FRAP; the results of which are presented in Table 4. Furthermore, the optimized condition was applied to the extraction process, and the TF content, TEAC, and FRAP were measured and tested. The optimal conditions that produced the maximum values for each factor were thus determined to be an ethanol concentration of 68.8%, a sample to solvent ratio of 1:10, an extraction temperature of 80 °C, and an extraction time of 13.64 h. These conditions exhibited a strong correlation between the predicted values and the experimental values, maximized the predictive potential for the fit of the model, and were, furthermore, shown to be the optimal conditions. We, therefore, recommend the adoption of these conditions in extraction protocols regarding DMLE to attain a maximum level of antioxidant activity of DMLE. RSM can be concluded to be a reliable and effective way to model and optimize extraction conditions. Similarly, Choi et al. studied the effect of reducing power on the antioxidant activity of DMLE and showed that an increase in reducing power with an increasing concentration of ethanol as the extraction solvent led to the highest reducing power at 80% ethanol extract, but a decreasing trend at 100% ethanol. This was attributed to the difference in the content of phenolic compounds in DMLE [28]. Another study on the correlation between TF content and the antioxidant potential of DMLE in each ethanol concentration reported that although TF content increased with increasing ethanol concentration and reached the highest level at 80% ethanol extract, it decreased at 100% ethanol. Moreover, physiological functionality, including the antioxidant activity, was shown to be proportional to the TF content [29]. The findings indicated that DML has the potential to be a good source of powerful antioxidant agents, and through associated antioxidant functions, can be an effective material for use in the prevention of various diseases caused by oxidative stress.

### 3.6. Toxicity of EtOH and Hepatoprotective Effects of DMLE

When ethanol is absorbed in the human body, it is metabolized and detoxified in the liver through the actions of ADH and ALDH to produce acetyl-CoA, which be used in the metabolic processes in the human body [30]. However, excessive alcohol consumption that overwhelms the metabolic and detoxifying processes mediated by ADH and ALDH, and is known to lead to a metabolic process via CYP2E1 that generates the ROS and induces liver injury [31]. The preventive effects of DMLE on alcohol-induced liver injury were, therefore, measured. The hepatoprotective effect of DMLE was investigated based on a previous study wherein human liver cancer cell line HepG2 cells were used and oxidative stress was induced with ethanol [5]. The result of the MTT assay showed that cytotoxicity was absent up to the DMLE concentration of 50 µg/mL, while cell survival rate began to fall from 100 µg/mL (Figure 4A). The cytotoxicity results of ethanol on HepG2 cells showed that the cell survival rate decreased with increasing ethanol concentration, whereby the rates associated with the ethanol concentrations of 100, 200, 400, 600, 800, and 1000 mM were 97%, 83%, 63%, 54%, 32%, and 26%, respectively (Figure 4B). Based on these findings, experiments were conducted using the concentration of DMLE that prevented cytotoxicity (12.5–50.0 µg/mL) and the concentration of ethanol that induced liver cell damage (400 mM). In addition, when the cells were treated with varying concentrations of DMLE for 12 h, followed by the treatment with ethanol for 24 h, the ethanol-treated group showed a decline in the cell survival rate to 60.34%. DMLE treatment was shown to have led to a cell survival of 72.87% at 12.5 μg/mL, 81.75% at 25 μg/mL, and 87.54% at 50 μg/mL (Figure 4C). These results provide support that DMLE has preventive effects on liver cell damage caused by ethanol.

### 3.7. DMLE Inhibited EtOH Induced Apoptosis and ROS Production

The activity of DMLE in the inhibition of apoptosis induction and ROS production caused by ethanol in liver cells was measured. HepG2 cells were pre-treated with DMLE for 24 h, then treated with ethanol for 24 h. The cells were stained using Annexin/PI and analyzed using FACS. The results showed that, in the ethanol-treated group, the number of apoptotic cells had increased to 39.25%, whereas in the DMLE-treated group, the apoptotic cells had decreased in a concentration-dependent manner. The number of cells was 31.75% in the group treated with 12.5 μg/mL DMLE; 25.98% in the group treated with 25 μg/mL; 21.54% in the group treated with 50 μg/mL, and all had a lower number of apoptotic cells after DMLE treatment (Figure 5A). To verify whether the apparent effect of ethanol on apoptosis is derived from ROS, the intracellular production of ROS after DCFH-DA was measured using FACS. The treatment of HepG2 cells with ethanol led to a steep increase in ROS; however, when the cells were pre-treated with DMLE prior to the ethanol treatment, the ROS production was markedly reduced (Figure 5B). These results indicated that, while HepG2 cells would show cell damage due to ROS-induced oxidative stress after ethanol treatment, the treatment with DMLE could inhibit ROS production and hence prevent cell damage caused by oxidative stress. When rat primary hepatocytes were treated with hot-water extracts of DML in a study on alcohol-induced cell damage, the DMLE was shown to have reduced the intracellular ROS and exerted a cytoprotective effect in the liver, which is consistent with the results in the present study [32].

### 3.8. Chemical Composition Analysis in DMLE

The functionality of various medicinal plant extracts originates from diverse polyphenolic compounds found in each extract. Notably, as reported by different studies, phenolic acid and flavonoid compounds that are abundantly found in plants exhibit the ability to prevent alcoholic liver injury [33,34]. This study thus investigated the compounds found in DMLE using HPLC Q-TOF MS analysis, so as to examine the active components that influence alcoholic liver injury. The HPLC UV chromatogram (280 nm) and total ion current chromatogram for DMLE are shown in Figure 6A. The molecular ion mass observed in the cation mode and the fragment ion mass observed in MS/MS analysis, as well as the compounds analyzed based on MS data, are presented in Figure 6B,C. The results of the HPLC Q-TOF MS analysis showed that the main compounds found in DMLE were caffeic quinic acid derivative and rutin. The peaks isolated after retention times of 7.6, 10.7, and 11.1 min displayed an identical UV absorption pattern with the maximum peak at 218 and 326 nm. The MS analysis in the cation mode produced a [M + H]^+^ of 355, which was determined to be an isomer of caffeic quinic acid, where caffeic acid is bonded to the hydroxyl group of quinic acid, based on identical molecular ions; the molecular ion peak at *m/z* 163.1 after the removal of H_2_O from caffeic acid; the peak at *m/z* 135.1 after the removal of CO from the peak at *m/z* 163.03. This pattern of molecular ion peaks agreed with the results of a previous study [35]. The peak at 19.9 min retention time exhibited maximum UV absorption peaks at 206, 256, and 353 nm, with molecular ion values of [M + H]^+^ and [M + Na] at *m/z* 611.15 and *m/z* 633.14, respectively. In addition, a molecular ion peak was found at *m/z* 465.10, which corresponds to rhamnose; the removal of *m/z* 146 from *m/z* 611.15. A peak was also found at *m/z* 303.04, which corresponds to glucose; the removal of *m/z* 162 from *m/z* 465.10. Based on these findings, we determined the compound to be a flavonoid glycoside, whereby a flavonoid (with a molecular mass of 302) was combined with rutinoside. The peak at *m/z* 303.04 after the MS/MS analysis also indicated that the flavonoid aglycon (with a molecular mass of 302) was, in fact, quercetin. These findings confirmed that the molecule was rutin, where rutinose is bound to quercetin, and such a molecular ion pattern was consistent with the results of a previous study [36].

The contents of chlorogenic acid and rutin in DMLE were quantified using the standard products and were found to be 12.56 and 32.75 mg, respectively, per 1 g of DMLE (Table 5). In a study by Hyun et al. [8] on the anticancer activities of premature and mature DMLE on Huh-7 liver cancer cells, the contents of chlorogenic acid and rutin in premature and mature DMLE were 4.32 ± 0.06 and 4.78 ± 0.04 mg/g and 10.46 ± 0.64 and 5.39 ± 0.22 mg/g, respectively. These findings were consistent with those of a study on the preventive effects of DMLE on gout, whereby the main components of the ethanol extract, whose powerful antioxidant function led to the anti-gout activity, were chlorogenic acid and rutin [37]. In another study on chlorogenic acid and rutin contents in *D. morbifera* leaves according to their different places of origin and extraction solvents, although the contents varied according to the place of origin, chlorogenic acid and rutin were still found to be the main components of DMLE, and the 80% ethanol extract condition exhibited the highest extract contents with the most outstanding antioxidant activity [28].

Chlorogenic acid is a phenolic acid derivative that is found in abundance in plants. The substance is contained in high concentrations in coffee and is known to exhibit myriads of functionalities, including antioxidant, anti-inflammatory, and anticancer activities, with preventive effects on alcoholic or non-alcoholic fatty liver disease [38]. Rutin, furthermore, is a flavonoid glycoside that is found in abundance in cereal grains such as buckwheat and oats, and based on its powerful antioxidant activity, it is known to exert protective effects against various liver diseases such as alcoholic liver injury [39,40]. DMLE in this study was also found to exhibit a protective effect against alcoholic liver injury, which has been determined to be attributed to the high chlorogenic acid and rutin contents in DMLE.

## 4. Conclusions

This study applied the RSM to optimize the extraction conditions for the production of DMLE, wherein the antioxidant activities of extracts such as the TF content, FRAP, and TEAC could be maximized. The factors that influence the extraction process, i.e., ethanol concentration, sample to solvent ratio, extraction temperature, and extraction time, were selected and analyzed based on the central composite design. The optimal extraction conditions for 1 g sample with 50 mL solvent were determined to be 70% ethanol, at 80 °C, for 14 h. The DMLE obtained using these optimal conditions contained chlorogenic acid and rutin, and exhibited an outstanding antioxidant function. This antioxidant function was found to inhibit the alcohol-induced formation of ROS inside cells, thus preventing liver cell damage. Further research, including an in vivo study, is required to identify the detailed mechanisms of DMLE against alcoholic liver injury. There are currently ongoing in vitro studies aiming to evaluate the biological activities and mechanisms of the bioactive compounds isolated from DMLE and to verify the bioactivity of DMLE. The findings of the present study suggest that DMLE may potentially be a useful therapeutic and preventive agent for alcoholic liver injury.

## Figures and Tables

**Figure 1 antioxidants-09-00120-f001:**
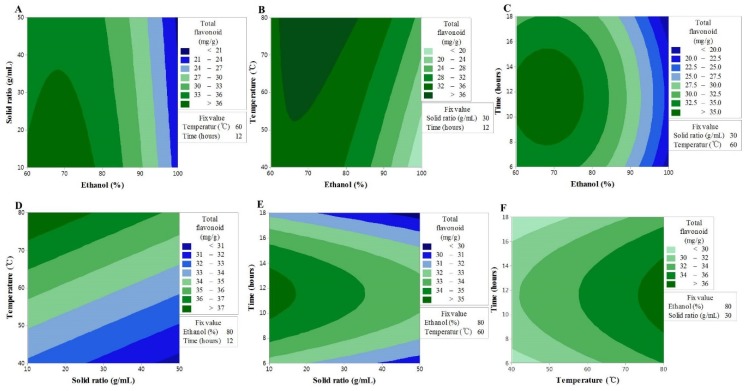
Contour plots showing the effect of ethanol concentration, solvent ratios, temperature, and time on total flavonoid content. (**A**) solid ratios (g/mL) vs. ethanol concentration (%); (**B**) temperature (°C) vs. ethanol concentration (%); (**C**) time (h) vs. US ethanol concentration (%); (**D**) temperature (°C) vs solid ratios (g/mL); (**E**) time (h) vs solid ratios (g/mL); (**F**) time (h) vs temperature (°C).

**Figure 2 antioxidants-09-00120-f002:**
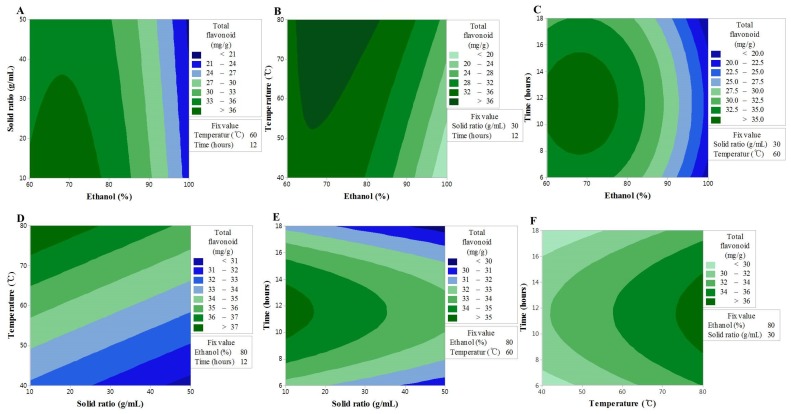
Contour plots showing the effect of ethanol concentration, solvent ratios, temperature, and time on Trolox equivalent antioxidant capacity. (**A**) solid ratios (g/mL) vs. ethanol concentration (%); (**B**) temperature (°C) vs. ethanol concentration (%); (**C**) time (h) vs. US ethanol concentration (%); (**D**) temperature (°C) vs. solid ratios (g/mL); (**E**) time (h) vs. solid ratios (g/mL); (**F**) time (h) vs. temperature (°C).

**Figure 3 antioxidants-09-00120-f003:**
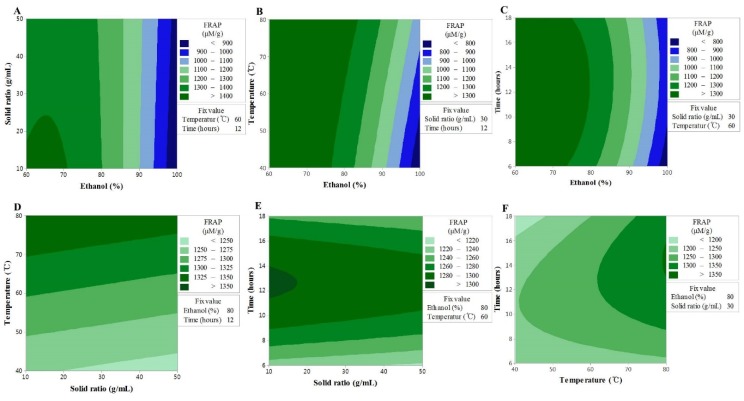
Contour plots showing the effect of ethanol concentration, solvent ratios, temperature, and time on ferric reducing antioxidant power. (**A**) solid ratios (g/mL) vs. ethanol concentration (%); (**B**) temperature (°C) vs. ethanol concentration (%); (**C**) time (h) vs. US ethanol concentration (%); (**D**) temperature (°C) vs. solid ratios (g/mL); (**E**) time (h) vs. solid ratios (g/mL); (**F**) time (h) vs. temperature (°C).

**Figure 4 antioxidants-09-00120-f004:**
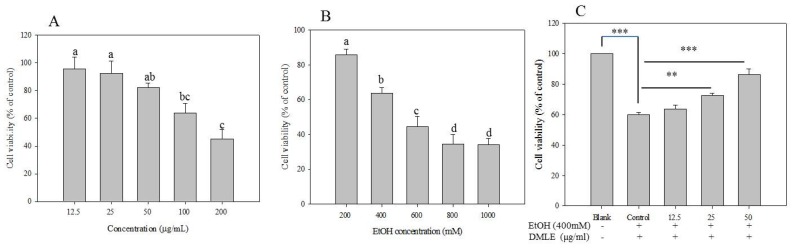
Cell viability (**A**), ethanol toxicity (**B**), and cytoprotective effect (**C**) of *Denropanax morbifera* leaf extracts. HepG2 cells exposed to deferent concentration DMLE and EtOH for 24 h. Cell viability measured by MTT assay. Data were reported as the percentage change in comparison to a blank group, which were assigned 100% viability. Data represent the mean ± S.E.M of three replicates. Different letters (a–d) indicated significant difference at *p* < 0.05. ** *p* < 0.01, *** *p* < 0.001.

**Figure 5 antioxidants-09-00120-f005:**
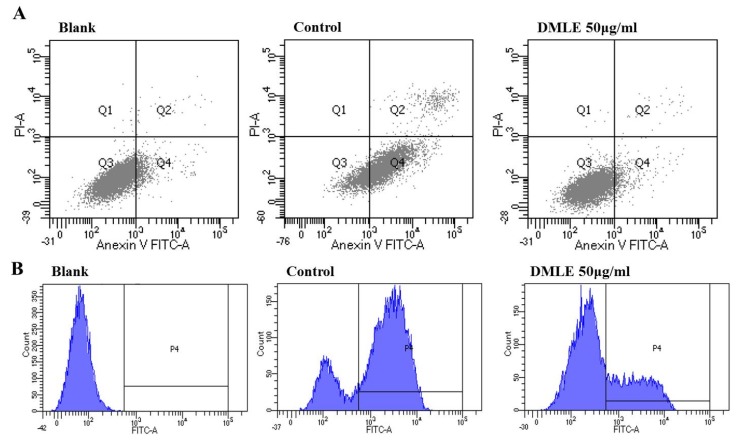
Effect of *D. morbifera* leaf extracts (DMLE) on EtOH induced HepG2 cell apoptosis and ROS production. (**A**) Apoptosis quantification of annexin V-FITC positive cells in the top (PI negative) and bottom (PI positive) right quadrant were indicated. (**B**) ROS induction in HepG2 cells was either treated with different concentrations of DMLE before EtOH treatment. ROS generation was measured by a flow cytometer. Cells were treated with different concentrations of DMLE for 1 h, and then exposed to ethanol (400 μM) for 24 h.

**Figure 6 antioxidants-09-00120-f006:**
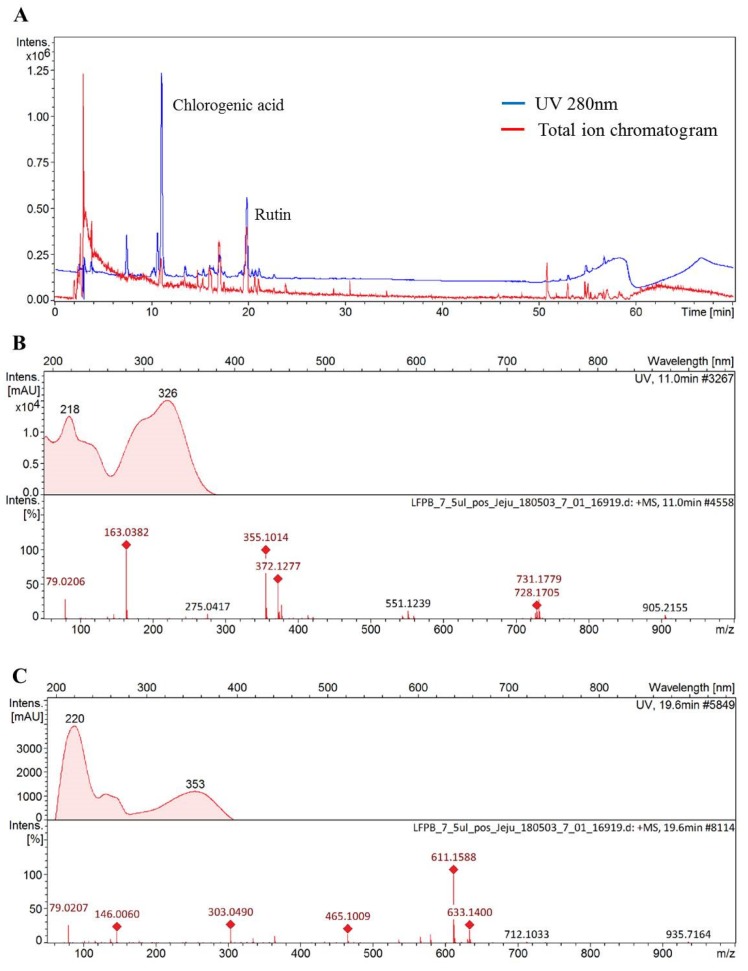
Chromatogram of *Dendropanax morbifera* leaf extract obtained at the optimum conditions at 280 nm (**A**), mass spectra for chlorogenic acid (**B**), and rutin (**C**). samples identified and confirmed using standard of reference.

**Table 1 antioxidants-09-00120-t001:** Coded settings for the process parameters for extraction according to a Central composition design design of RSM.

Symbols	Experiment Factors	Coded Levels		
		−1	0	1
X_1_	Ethanol concentration (%)	60	80	100
X_2_	Sample to solvent ratio (g/mL)	1:10	1:30	1:50
X_3_	Extraction temperature(°C)	40	60	80
X_4_	Extraction time (h)	6	12	18

**Table 2 antioxidants-09-00120-t002:** Central composition design and response values of the total flavonoid contents and antioxidant activity of the *D**, morbifera* leaf extracts.

Run	Independent Variables				Dependent Variables (Response)		
	X_1_	X_2_	X_3_	X_4_	TF	TEAC	FRAP
1	80	30	60	12	69.20	35.48	148.19
2	80	30	60	12	68.72	35.41	147.27
3	100	10	40	18	77.95	14.99	318.17
4	100	10	40	6	62.61	12.65	374.02
5	60	50	40	6	81.35	32.41	140.26
6	100	10	80	6	42.33	26.47	261.75
7	60	10	40	18	85.38	32.07	148.45
8	60	50	80	6	63.51	31.67	152.85
9	60	10	40	6	59.28	37.88	142.56
10	80	30	60	12	71.36	36.59	144.04
11	100	50	80	6	64.90	24.07	254.04
12	100	50	40	6	75.24	9.48	304.61
13	60	50	40	18	64.21	27.45	147.88
14	100	10	80	18	39.63	19.41	249.73
15	100	50	80	18	45.11	23.72	236.17
16	60	10	80	6	61.98	33.98	144.54
17	60	50	80	18	76.08	30.96	147.92
18	100	50	40	18	53.45	13.88	318.21
19	60	10	80	18	74.34	33.50	144.02
20	60	30	60	12	53.79	32.19	136.05
21	100	30	60	12	74.41	22.72	286.79
22	80	10	60	12	48.17	34.36	149.15
23	80	50	60	12	64.41	32.10	148.96
24	80	30	40	12	46.36	31.68	155.42
25	80	30	80	12	73.86	34.66	158.48
26	80	30	60	6	56.64	27.67	159.16
27	80	30	60	18	64.83	32.44	150.69
28	80	30	60	12	70.52	34.80	147.34
29	80	30	60	12	70.87	35.36	145.82
30	80	30	60	12	69.97	34.40	146.98

X_1_ = ethanol concentration (%), X_2_ = sample to solvent ratio (ml/g), X_3_ = extraction temperature (°C), X_4_ = extraction time (h), TF = total flavonoid content (mg QE/1g extraxts), TEAC = Trolox equivalent antioxidant capacity (mM Trolox/1 g extracts), FRAP = ferric reducing antioxidant power (mM Fe^2+^/1 g extracts).

**Table 3 antioxidants-09-00120-t003:** Analysis of variance of RSM for the for total flavonoids, TEAC and FRAP values.

Response Variable	Factors	Degree of Freedom	Mean of Square	F Value	*p-*Value
TF (R)	Model	7	234.43	41.45	0.001
	Linear	4	251.48	44.46	0.001
	X_1_	1	863.84	152.73	0.001
	X_2_	1	21.29	3.76	0.065
	X_3_	1	117.35	20.75	0.001
	X_4_	1	3.43	0.61	0.444
	Quadratic	2	261.45	46.23	0.001
	X_1_^2^	1	118.31	20.92	0.001
	X_4_^2^	1	36.68	6.49	0.018
	Cross product	1	112.20	19.84	0.001
	X_13_	1	112.20	19.84	0.001
	Error	1	5.66		
	Lack of fit	1	7.16	13.00	0.005
	*R^2^*	0.929			
	*R^2^* _adj_	0.846			
TEAC	Model	9	6535.0	46.97	0.001
	Linear	4	11,513.1	87.51	0.001
	X_1_	1	39,667.4	301.52	0.001
	X_2_	1	1165.8	8.86	0.007
	X_3_	1	4671.2	35.51	0.001
	X_4_	1	547.9	4.16	0.055
	Quadratic	1	6539.5	49.71	0.001
	X_1_^2^	1	6539.5	49.71	0.001
	Cross product	4	1555.8	11.83	0.001
	X_13_	1	2057.5	15.64	0.001
	X_14_	1	1643.8	12.49	0.002
	X_23_	1	975.0	7.41	0.013
	X_3__4_	1	1546.9	11.76	0.003
	Error	20	131.6		
	Lack of fit	15	173.3	27.00	0.001
	*R^2^*	0.957			
	*R^2^* _adj_	0.862			
FRAP	Model	10	190,482	60.56	0.001
	Linear	4	345,311	109.79	0.001
	X_1_	1	1,334,388	424.28	0.001
	X_2_	1	955	0.30	0.588
	X_3_	1	42,619	13.55	0.002
	X_4_	1	3283	1.04	0.320
	Quadratic	2	202,258	64.31	0.001
	X_1_^2^	1	126,149	40.11	0.001
	X_4_^2^	1	11,752	3.74	0.068
	Cross product	4	29,764	9.46	0.001
	X_12_	1	15,868	5.05	0.037
	X_13_	1	71,508	22.74	0.001
	X_14_	1	12,704	4.04	0.059
	X_34_	1	18,978	6.03	0.024
	Error	19	3145		
	Lack of fit	14	4190	19.08	0.002
	*R^2^*	0.969			
	*R^2^* _adj_	0.909			

X_1_ = ethanol concentration (%), X_2_ = sample to solvent ratio (ml/g), X_3_ = extraction temperature (°C), X_4_ = extraction time (h), TF = total flavonoid content (mg QE/1g extraxts), TEAC = Trolox equivalent antioxidant capacity (mM Trolox/1 g extracts), FRAP = ferric reducing antioxidant power (mM Fe^2+^/1 g extracts).

**Table 4 antioxidants-09-00120-t004:** Experimental data of the validation of predicted values at optimal extraction conditions.

Dependent Variable	Predict Value	Experimental Value
TF	37.40	35.30 ± 2.06
FRAP	1387.8	1338.60 ± 62.21
TEAC	288.99	282.43 ± 6.90

TF = total flavonoid content (mg QE/1 g extracts), TEAC = Trolox equivalent antioxidant capacity (mM Trolox/1 g extracts), FRAP = ferric reducing antioxidant power (mM Fe^2+^/1 g extracts).

**Table 5 antioxidants-09-00120-t005:** Chlorogenic acid and Rutin contents of *Dendropanax morbifera* leaf extract under optimal extraction conditions. Data represent the mean ± S.E.M of three replicates.

Compound	Linear	R^2^	Concentration (mg/g Extracts)
Chlorogenic acid	*Y* = 27288.97χ + 980.54	0.9998	34.33 ± 0.16
Rutin	*Y* = 0.0062χ + 0.459	0.983	91.93 ± 0.56

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
