# Peer review of "Dendropanax morbifera Leaf Polyphenolic Compounds: Optimal Extraction Using the Response Surface Method and Their Protective Effects against Alcohol-Induced Liver Damage"

_antioxidants, 2020, doi:10.3390/antiox9020120_

Round 1
Reviewer 1 Report
The manuscript presented for review Dendropanax morbifera leaf polyphenolic 3 compounds: Optimal extraction using the response 4 surface method and their protective effects against 5 alcohol-induced liver damage is very interesting and addresses many research and problematic threads
The authors of the work in the first section (Introduction) in an accessible and very careful way describe the newest knowledge in experiment area. The literature used is very new and covers the years 2001-2016. That is why I think the problem is serious and the authors of the work approached it very well. The collected research material is beyond doubt. Please add some information indicated in the detailed points.
The methods used to extract polyphenolic compounds and the methods of analysis are very modern and do not raise any doubts.
Section Result and discussion is a very good described. As well obtained results are compare to those from literature review.
I think the work is very valuable and should be published in the Journal.
My detailed points:
Line 42-43, page 2: Please use italics for Latin species names
line 84, page 2: please add drying time of leaves in room temperature
line 84, page 5: please add gradient program for all time and fazes (A and B)
Reviewer 2 Report
Dear editor and authors,
In general the study is interesting, but the manuscript needs further revision. There are some inconsistencies in the models and in the results, considering the information provided in Table 3 and Figure 6. For that reason, I recommend rejection. Moreover, the material and methods part (2.10 and 2.11) should be improved.
Other comments:
Please, confirm if the accepted name is Dendropanax morbiferusLév. (see http://www.theplantlist.org/tpl1.1/record/kew-60088) Species names should be in cursive letter. Please, redefine the objective to indicate what response variables are going to be studied. Line 83, delete the extra “,”. Line 91, rephrase to be more accurate. Line 95, instead of stages, write levels Revise grammatical errors in Table 1 and legend. Revise equation 1 (the box does not have good quality) Line 120, define SD Line 145. From the latter assay in section 2.8? please, clarify. Line 155. From the latter assay in section 2.8? please, clarify. Section 2.10. Define the abbreviations used. Revise paragraph 160-163. Add characteristics of the column (C18??), the gradient used, the injection volume, and the MS/MS parameters. m/z should be with cursive letter. Define abbreviations in line 171. Line 173, “from YMC Co., Ltd (Tokyo, Japan)” is not necessary. After line 173, add the gradient, injection volume, and the wavelength (or MS peak) used for quantification. Revise grammatical errors in Table 2 and Table 4 (e.g. missing spaces, etc). Also, “hours” should be “h” Table 3, please, instead of “-“, add the corresponding value and the statistical significance. Please, be concise in sections 3.1.-3.4. Firstly, this means that you can delete all repetitive information (from lines 186 to 210) in section 3.1., which can be read in the other sections or vice versa. Moreover, the information in the text seems not correct considering the current information provided in Table 3.Moreover, what did you mean with small standard deviation?? Did you repeat the extractions? There is no data about it.
Secondly, since the information should be in Table 3, you can be more concise in 3.2 to 3.4.
Line 228-230, the information is not clear. This means that p-value for lack-of-fit should be higher than 0.05 to be an acceptable model. In general, the table 3 information does not correspond with the text. Also, bear in mind that if a linear or the quadratic effect of a facto is not significant, this interaction with other factor should not be considered in the final model. The terms no significant should also be removed from the equation models. Thus, table 3 should be revised as well as the models. Figure 6 does not correspond to the optimal conditions provided by the model (based on the legend). Line 412-414, revise the information. Bear in mind that you comment only two compounds, but the UV spectra and the TIC show other minor compounds. Instead of cation, you can use positive ionization mode. Nonetheless, you mention that the analyses were conducted in the negative ion mode in the experimental part. Please, correct the right information. A table, at least as supporting information, can be added with the information (UV, MS, MS/MS data) of all the compounds characterized. Moreover, the isomers of chlorogenic can be also quantified with the curve of chlorogenic acid to see more or less the concentration range. Table 5. Clarify how the compounds were quantified. With UV? Bear in mind that the slope are quite different between compounds.Author Response
Response to Reviewer 2 Comments
Point 1: Please, confirm if the accepted name is Dendropanax morbiferusLév. Species names should be in cursive letter.
Response 1: scientific name is correct.
Point 2: redefine the objective to indicate what response variables are going to be studied.
Response 2: The purpose of this study was to optimize the extraction parameters to produce extracts with good antioxidant activity. Among the antioxidant variables, TEAC and FRAP assays were used as response variables, and flavonoid content, known as antioxidant, was used to explain the relationship with antioxidant activity. In addition, an extract with excellent antioxidant activity was used to study the inhibition of liver damage to free radicals. Consequently, the purpose of this study is clearly explained in the manuscripts.
Point 3: Line 83, delete the extra “,”
Response 3: Deleted extra “,”
Point 4: Line 91, rephrase to be more accurate.
Response 4: Change to RSM Experimental Design
Point 5: Line 95, instead of stages, write levels.
Response 5: Marked as a level
Point 6: Revise grammatical errors in Table 1 and legend. Revise equation 1 (the box does not have good quality).
Response 6: Table legends equation change
Point 7: Line 120, define SD.
Response 7: Added SD definition.
Point 8: Line 145 From the latter assay in section 2.8? please, clarify.
Response 8: Change to Apoptosis Analysis using Flow Cytometry
Point 9: Line 155. From the latter assay in section 2.8? please, clarify.
Response 9: Change to cellular ROS Measurement using Flow Cytometry
Point 10: Section 2.10. Define the abbreviations used. Revise paragraph 160-163. Add characteristics of the column (C18??), the gradient used, the injection volume, and the MS/MS parameters. m/z should be with cursive letter. Define abbreviations in line 171. Line 173, “from YMC Co., Ltd (Tokyo, Japan)” is not necessary. After line 173, add the gradient, injection volume, and the wavelength (or MS peak) used for quantification.
Response 10: Modified section 2.10 and 2.11
Point 11: Revise grammatical errors in Table 2 and Table 4 (e.g. missing spaces, etc). Also, Response 11: “hours” should be “h”. Table 2 and 4 legends changed.
Point 12: Table 3, please, instead of “-“, add the corresponding value and the statistical significance.
Response 12: Added ‘-‘ is nonsignificant
Point 13: Please, be concise in sections 3.1.-3.4. Firstly, this means that you can delete all repetitive information (from lines 186 to 210) in section 3.1., which can be read in the other sections or vice versa. Moreover, the information in the text seems not correct considering the current information provided in Table 3. Lines 186 to 210 are summaries of response Response 13: surface analysis and are considered necessary for manuscript.
Point 14: Moreover, what did you mean with small standard deviation? Did you repeat the extractions? There is no data about it. In the RSM analysis, the error is calculated by Response 14: comparing the midpoint test for minimizing the experiment.
Point 15: Secondly, since the information should be in Table 3, you can be more concise in 3.2 to 3.4.
Line 228-230, the information is not clear. This means that p-value for lack-of-fit should be higher than 0.05 to be an acceptable model. In general, the table 3 information does not correspond with the text. Also, bear in mind that if a linear or the quadratic effect of a facto is not significant, this interaction with other factor should not be considered in the final model. The terms no significant should also be removed from the equation models. Thus, table 3 should be revised as well as the models.
Response 15: Lack of fit means that it is difficult to trust the regression model because the variation produced because the polynomial regression model obtained through the experiment is not suitable to explain the response is so large that it cannot be ignored. Statistics for significance tests use F-values. This is because the mean square of lack of fit is divided by the pure error mean square, which follows the F-distribution.
The reason why the regression model obtained through the experiment is not suitable for explaining the response is that most of the regression model orders are not suitable or the important experimental factors are missing from the regression model.
In addition, even though the linear and quadratic effects of factors in the optimal model are not significant, they should be included in the final model because they consider the interactions. Therefore, I think it is exactly expressed in manuscript.
Point 16: Line 412-414, revise the information. Bear in mind that you comment only two compounds, but the UV spectra and the TIC show other minor compounds. Instead of cation, you can use positive ionization mode. Nonetheless, you mention that the analyses were conducted in the negative ion mode in the experimental part. Please, correct the right information. A table, at least as supporting information, can be added with the information (UV, MS, MS/MS data) of all the compounds characterized. Moreover, the isomers of chlorogenic can be also quantified with the curve of chlorogenic acid to see more or less the concentration range.
Table 5. Clarify how the compounds were quantified. With UV? Bear in mind that the slope are quite different between compounds.
Response 16: Mass spectrometry was carried out in cation mode, and mass spectrometry confirmed that chlorogenic acid and ruin were the main components. Therefore, each standard was quantified using HPLC.

Reviewer 3 Report
In the study, the method for preparation of Dendropanax morbifera leaf extracts (DMLE) was optimized using the response surface method (RSM) with four independent and three dependent variables, and the effect of DMLE on liver cells treated with ethanol was examined. The optimum condition for extraction was successfully determined, and DMLE prepared with the optimized method prevented the liver injury caused with ethanol by scavenging the reactive oxygen species. Additionally, high concentrations of chlorogenic acid and rutin in the DMLE were confirmed.
I think the quality of the manuscript is satisfactory. However there are several issues to be improved for better understanding.
All the numbers in the manuscript should be round to three-significant digits at maximum including regression coefficients. The information in LC analyses is not enough. In the section 2.10, there is no information about mobile phases. In the section 2.11, there is no information about gradient program together wit flow rate. Section 3.5: Since the aim of the study was optimize the extraction method, the authors should compare the results with the methods previously reported. Otherwise, it is difficult to judge whether the established method was really optimal. Section 3.8: The result of LC-MS analysis should be explained in an accurate way. For example, I couldn’t follow the description about the peaks at RT 7.6 and 10.7 because no data was shown for those peaks. Moreover the numbers in the manuscript and figures are not coincident. Lastly, if the authors identified the compound by comparing the data with those of authentic compounds, data of authentic compounds should be shown. I don’t think that table 5 is necessary, since no information about Y and x is provided.Additionally, there are several technical incorrectnesses as follows.
Line 42–46: All scientific names should be in Italic. Line 73–77: Starting with capital letters for spelling of compounds is inappropriate. Line 93: “n” in “Xn” should be subscript. Line 104: “o” in “βo” should be “0” in subscript. Table 1: All the numbers in “X1, X2, X3 and X4” should be subscripts. Line 161: “DAD detector” should be “a diode-array detector (DAD)”. Line 164, 165: “a diode-array detector” should be “DAD”. Line 166: “m” and “z” in “m/z” should be in Italic. Table 3: “X0” should be “β0”. Line 272: “TA” should be “TF”.Author Response
Response to Reviewer 3 Comments
Point 1: Line 42-43, page 2: Please use italics for Latin species names
Response 1: Change to italics (Line 42-43)
Point 2: All the numbers in the manuscript should be round to three-significant digits at maximum including regression coefficients.
Response 2: Represents within three-significant digits.
Point 3: The information in LC analyses is not enough. In the section 2.10, there is no information about mobile phases. In the section 2.11, there is no information about gradient program together with flow rate.
Response 3: Adding HLPC solvent and gradient conditions.
Point 4: Section 3.8: The result of LC-MS analysis should be explained in an accurate way. For example, I couldn’t follow the description about the peaks at RT 7.6 and 10.7 because no data was shown for those peaks. Moreover the numbers in the manuscript and figures are not coincident. Lastly, if the authors identified the compound by comparing the data with those of authentic compounds, data of authentic compounds should be shown. I don’t think that table 5 is necessary, since no information about Y and x is provided.
Response 4: Peaks of 7.6 and 10.7 minutes are isomers in which caffeic acid is bound to qunic acid. Chlorogenic acid peak is appeared 11.1 minutes. The quantification of chlorogenic acid was calculated using a peak of 11.1 minutes.
Point 5: Point 2: Line 42–46: All scientific names should be in Italic.
Response 5: Represent to italics
Point 6: Line 73–77: Starting with capital letters for spelling of compounds is inappropriate. Response 6: Modified and displayed in manuscript. Point 7: Line 93: “n” in “Xn” should be subscript. Line 104: “o” in “βo” should be “0” in subscript. Response 7: Modified and displayed in manuscript.Point 8: Table 1: All the numbers in “X1, X2, X3 and X4” should be subscripts. Response 8: Modified and displayed in manuscript. Point 9: Line 161: “DAD detector” should be “a diode-array detector (DAD)”. Line 164, 165: “a diode-array detector” should be “DAD” .Response 9: Modified and displayed in manuscript.
Point 10: Line 166: “m” and “z” in “m/z” should be in Italic.
Response 10: Represent to italics
Point 11: Table 3: “X0” should be “β0”. Response 11: Modified and displayed in manuscript. Point 12: Line 272: “TA” should be “TF”. Response 12: Modified and displayed in manuscript.

Reviewer 4 Report
The manuscript presented for publication in Antioxidants Journal, is very interesting in respect of the methods and finding optimal conditions for
bioactive compounds extraction. The design using independent variables on dependent variables of interest, to predict the optimal conditions for the
analysis.
Minor observation from my review:
Provide the equation and r2 for quercetin (line 116), Trolox (line 1234) and FeSO4 (line 130).
Line 176: where is the gradient of the HPLC method?
Overall, my recommendation is Minor revision
Author Response
Response to Reviewer 4 Comments
Point 1: Provide the equation and r2 for quercetin (line 116), Trolox (line 1234) and FeSO4 (line 130).
Response 1: The first order regression equation and R2 were added in the experimental analysis method.
Point 2: Line 176: where is the gradient of the HPLC method? Adding HLPC solvent and gradient conditions.
Response 2: Adding HLPC solvent and gradient conditions.

Round 2
Reviewer 2 Report
Dear authors and reviewers,
I think that the manuscript is interesting as commented before. Although the authors revised some parts, some answers were vague.
I still consider that the table 3 does not conform to what is indicated in the text; the data provided in table 3 should be revised (including what terms are significant, including footnotes) and the models should be revised (e.g. F value (model) are not significant in some cases).
The table with the calibration curves of chlorogenic acid and rutin need to add the range. Some comments about the slopes are expected, since that of chlorogenic is very low.
The text needs grammatical revision as well as format edition.
So, I recommed that authors improve the manuscript before resubmission.
Author Response
Table 3 is changed to the analysis of variance (ANOVA).
The ANOVA was used to evaluate the significance of the quadratic polynomial models. For each term in the models, a large F-value and a small P-value would imply a more significant effect on the respective response variable. Bases on the F-value and p-values for the experimental results, this model is considered appropriate.
To verify the adequacy of a model, the coefficient of determination (R2), lack of fit and R2adj tests were typically used. R2 represents a percentage of the variables that can be explained by the model. Commonly, the higher R2 not only represents the majority of the variables that can be explained by the model, but also represents that the experimental data are very consistent with the second-order polynomial equation.